# Simultaneous Estimation of Vehicle Sideslip and Roll Angles Using an Event-Triggered-Based IoT Architecture

Fernando Viadero-Monasterio [1,*,†], Javier García [2,†], Miguel Meléndez-Useros [1], Manuel Jiménez-Salas [1], Beatriz López Boada [1] and María Jesús López Boada [1]

1   Mechanical Engineering Department, Advanced Vehicle Dynamics and Mechatronic Systems (VEDYMEC), Universidad Carlos III de Madrid, Avda. de la Universidad 30, 28911 Leganés, Spain; mmelende@ing.uc3m.es (M.M.-U.); manuejim@ing.uc3m.es (M.J.-S.); bboada@ing.uc3m.es (B.L.B.); mjboada@ing.uc3m.es (M.J.L.B.)

2   Computer Science and Software Engineering Department, Universidad Carlos III de Madrid, Avda. de la Universidad 30, 28911 Leganés, Spain; jgarciag@inf.uc3m.es

*   Correspondence: fviadero@ing.uc3m.es

†   These authors contributed equally to this work.

**Abstract:**  In recent years, there has been a significant integration of advanced technology into the automotive industry, aimed primarily at enhancing safety and ride comfort. While a notable proportion of these driver-assist systems focuses on skid prevention, insufficient attention has been paid to addressing other crucial scenarios, such as rollovers. The accurate estimation of slip and roll angles plays a vital role in ensuring vehicle control and safety, making these parameters essential, especially with the rise of modern technologies that incorporate networked communication and distributed computing. Furthermore, there exists a lag in the transmission of information between the various vehicle systems, including sensors, actuators, and controllers. This paper outlines the design of an IoT architecture that accurately estimates the sideslip angle and roll angle of a vehicle, while addressing network transmission delays with a networked control system and an event-triggered communication scheme. Experimental results are presented to validate the performance of the IoT architecture proposed. The event-triggered scheme of the IoT solution is used to decrease data transmission and prevent network overload.

**Keywords:** roll angle estimation; sideslip angle estimation; IoT; event-triggering





## 1. Introduction

Traffic accidents have a major socio-economic impact as they lead to both the loss of human lives and huge economic costs. According to global statistics, passengers in a rollover accident are 10 times more likely to die than those in a non-rollover accident, and this situation represents 33% of all passenger car crashes [1–3]. One line of action to prevent these accidents focuses on research into and the improvement of roll stability control systems, which are relevant for heavy vehicles [4–7]. Additionally, skid prevention systems based on torque distribution, such as differential braking, are a growing research trend for vehicles driving in slippery conditions [8–10].

Currently, vehicles incorporate control systems to improve vehicle stability and handling. These systems rely on knowledge of the vehicle's dynamics, either by measuring its state directly from sensors or by estimating those states through observer models [11–13]. The major obstacle in implementing these systems is the trade-off between processing capacity and the economic cost of the sensors. Sensors that provide direct measurements of the major parameters related to vehicle dynamics (pitch, roll angle, and sideslip) are expensive, and their incorporation into vehicles would increase production costs considerably [14–16]. As an example, inertial measurement unit (IMU) sensors, which cost around

EUR 3000, directly measure vehicle yaw, roll, and pitch rates using gyroscopes and the effect of Coriolis acceleration. However, sideslip and roll angles are still not measurable from IMUs. To overcome this problem, it is possible to install a dual-antenna GPS; however, since the price of this device exceeds EUR 13,000, it cannot be considered for series-production vehicles [1].

State estimation is currently a fundamental research topic in the field of automated driving systems (ADS) [17–19]. The aim of the filtering or estimation problem is to estimate the unknown states of a system from the information provided by the sensor signals. There are mathematical procedures that allow certain values to be estimated from data from other lower-cost sensors, such as a global positioning system (GPS) or an IMU. This form of approximation avoids the need to incorporate high-cost sensors but introduces delays derived from communications between components and the mathematical operations necessary to produce the different estimates from the data.

There are a wide range of studies that attempt to design estimation systems for states that cannot be measured. In order to estimate the different angles related to vehicle dynamics, processes such as the following are carried out:

- Using the velocities and accelerations of the axes X, Z;
- Merging the lateral accelerations and the roll rate [20];
- Combining the data obtained using a gyroscope and an inertial angle sensor [21];
- Operating using the values obtained using a six-dimensional IMU [22];
- Using a low-cost GPS together with the values from sensors located on the wheels [23].

An alternative solution is to provide reliable estimates from other sensors that are already included in series-production vehicles. This can be achieved through the Internet of Things (IoT), which consists of interconnecting different devices so that they can communicate with each other and perform different tasks, such as gathering information from different sensors or processing the data. Several typical requirements for IoT architectures are stated below:

1. The sampling rate of the sensors must be high in order to ensure that information is received fast enough to act in case of a particular event;
2. It is necessary to obtain information from low-cost sensors or devices already included in series-production automobiles in order to ensure the vehicle prices are reasonable;
3. Whenever data cannot be obtained directly, they must be estimated from direct measurements;
4. The elapsed time to perform computational tasks should be as short as possible in order to guarantee a rapid response to different events;
5. The developed system must have a low power consumption in order not to exhaust the vehicle resources.

With the aim of reducing the sampling rate and power requirements without compromising the accuracy of the results obtained, an alternative approach is to implement an event-triggering mechanism in order to transmit only relevant information to the system instead of all the data captured by the sensors [24–28]. This improves the system performance by reducing the number of data packets that need to be processed. In contrast to time-triggered data transmission, numerous studies have focused on the design of event-triggered systems [29–33]. In [29,30,33], event-triggered dynamic output feedback controllers for NCS are proposed. In [34], an event-triggering mechanism is used for distributed networked control systems considering network faults and communication delays. In [35], an adaptive event-triggering condition is designed while ensuring that the Zeno behavior is avoided. In [36,37], an active suspension is controlled through an event-triggered $\mathcal{H}_\infty$ controller in a networked control system with communication delays. Nevertheless, a significant portion of these results are based on simulations, making it difficult to find hard experimental evidence that validates the effectiveness of event-driven systems.

Effectively designed and developed software components that apply these techniques are crucial to meet the strict real-time restrictions when embedding these components in low-cost devices such as the Raspberry Pi 3 Model B or Intel Edison [38]. To reduce

computation times, low-level languages such as C++ must be employed for direct control over the hardware and to optimize computational time. However, it is not trivial to reduce the delays that may occur during data transmission through the network since these will be determined by the vehicle's network and the number of packets that are in circulation [39]. In addition, there is the possibility of network flooding and loss of information if the sensor sampling rate is higher than the packet processing capabilities.

For all the above reasons, the objective of this work was to design a system that allows for the estimation of vehicle sideslip and roll angles, with the information measured by low-cost sensors, while reducing the in-vehicle network usage through an event-triggering mechanism. In addition, code was developed to allow computers to communicate with each other in order to simulate a distributed system similar to one that would be found in a real vehicle. The novelty reflected in this work is the analysis of the vehicle roll, pitch, and yaw angle estimation using an event-triggering-based solution and low-cost devices under high dynamic conditions to avoid in-vehicle network saturation. The estimation outcomes were compared with the measurements acquired by a high-end professional device (VBOX from Racelogic), which were used as the ground truth.

This paper is structured as follows. In Section 2, the methodology is presented, including a description of the experimental testbed design, the experiments, and the data gathering and analysis techniques. The experimental results and discussion are presented in Section 3. In Section 4, the conclusions are outlined.

## 2. Research Approach

The purpose of this work was to develop an IoT solution to estimate both vehicle roll and sideslip angles by implementing event-triggering principles to reduce data traffic between different low-cost sensors and avoid vehicle network overload. Therefore, the hypothesis to be tested was whether it is feasible for an IoT system to approximate vehicle roll and sideslip angles in real time without overloading the intra-vehicular network.

The perspective of the hardware and communications on board the vehicle is presented in Section 2.1. The fundamentals of the event-triggered observer investigated in the scope of this research can be found in Section 2.2. The perspective of the software developed is explained in Section 2.3. Finally, information concerning the experiments and data gathering are provided in Sections 2.4 and 2.5, respectively.

### 2.1. IoT Hardware and Communications Perspective

To implement the solution, an IoT architecture was proposed in which the different components necessary for estimating roll and sideslip angles are integrated. There are various types of components involved, ranging from those facilitating the transfer of data to those performing the required calculations for output variable estimation. These are coupled with the sensors used for input parameter, acquisition enabling roll and slip angle estimations.

In order to enable communication between components, a local Gigabit Ethernet network was utilized. A switch was included to allow inter-connectivity between the different computers in the local network. Network cables with an RJ45 connector were also included to connect the different computers to the switch. This switch was connected to a 4G router by means of another local network cable, so that all computers connected to the local network had access to the Internet.

It is important to note that all the above local network components needed to support Gigabit Ethernet connections (IEEE 802.3z standard [40]). This technology was chosen because it ensures the data network had a sufficiently large bandwidth to support a high throughput of packets, while also having a latency that is lower than in 100 MBps networks. Thus, delays lower than 1 ms were achieved in a stable manner for the local vehicle network. Low latency is essential to achieving a correct estimation, since a network latency over a certain threshold makes the correct estimation of sideslip and roll angles difficult.

Two Pi 3b + model Raspberries, with 4-core processors and 1 GB of RAM, enabled computational processes within this system. The first received the information from the vehicle's sensors and executed the event-triggering algorithm, which decided which values obtained from the sensors should be sent to the network and which should be discarded. The information sent over the network was significantly reduced through this process, which, in turn, reduced the traffic on it. The decision as to which values to neglect was made based on their utility in estimating roll and slip angles. The second Raspberry executed the sideslip and roll angle observer; this received the data filtered by the event-triggering process at the other end of the network and implemented the estimator, which used these data to obtain the roll and sideslip angles.

Concerning vehicle measurements, three different sensors were mounted in order to obtain the necessary measurements for the proposed estimator. The first Raspberry comprised a low-cost IMU together with a GPS (GPS-PIE Gmm slice 9DOF IMU). The IMU provided the vehicle's roll and yaw rates in a range of 2000°/s, while the longitudinal velocity of the vehicle was measured from the GPS sensor in a range of 515 m/s. The steering wheel angle was measured through a Kistler steering wheel sensor. Moreover, three high-cost sensors with sampling frequencies of 100 Hz were mounted in order to compare the ground truth and estimated data:

- Racelogic IMU04. This sensor measures pitch, roll, and yaw rates using three rate gyroscopes. Moreover, it obtains the accelerations in the x, y, and z directions via three accelerometers with a ±20 g linear acceleration range and a 0.00004 g acceleration resolution, and a ±450°/s angular rate range with a 0.00085°/s resolution;
- Racelogic VBOX 3i Dual Antenna. Depending on the mounting configuration, this sensor can be used to measure the roll, pitch, and sideslip angles, and determine vehicle positioning with a 1 cm distance resolution, a 0.01 km/h velocity resolution, less than a 0.2° sideslip angle RMS accuracy, and a 0.14° roll angle RMS accuracy;
- Kistler steering wheel DTI sensor. This sensor can measure the steering wheel angle and torque with the following specifications: a ±250 Nm steering moment range, a ±1250° steering angle range, a maximum 2000°/s steering speed, and a 0.015° steering angle resolution.

The estimated output values and localization data were sent to the cloud through the 4G Router.

### 2.2. Event-Triggered Vehicle Sideslip and Roll Angles $\mathcal{H}_\infty$ Observer

The estimation problem is explored in this section. For the chosen vehicle model, it is assumed that the vehicle sprung mass rotates around the roll center. Moreover, the vehicle has three degrees of freedom: the sideslip angle ($\beta$), yaw rate ($r$), and roll angle ($\phi$). Vehicle parameters are presented in Table 1. The equations of lateral motion of the vehicle are [24]:

$$
\begin{aligned}
\dot{\beta} &= -\frac{I_{eq}C_0}{I_x m v_x}\beta - \left(1 + \frac{I_{eq}C_1}{I_x m v_x^2}\right)r + \frac{h(mgh - k_r)}{I_x v_x}\phi - \frac{hb_r}{I_x v_x}\dot{\phi} + \frac{I_{eq}C_{\alpha f}}{I_x m v_x}\delta - \frac{g}{v_x}\phi_r \\
\dot{r} &= -\frac{C_1}{I_z}\beta - \frac{g}{v_x}\phi_r + \frac{aC_{\alpha f}}{I_z}\delta \\
\ddot{\phi} &= -\frac{C_0 h}{I_x}\beta - \left(1 + \frac{I_{eq}C_1}{I_x m v_x^2}\right)r + \frac{mgh - k_r}{I_x}\phi - \frac{b_r}{I_x}\dot{\phi} + \frac{C_{\alpha f}h}{I_x}\delta - \dot{p}_f
\end{aligned}
\tag{1}
$$

where

$$
\begin{aligned}
C_0 &= C_{\alpha f} + C_{\alpha r} \\
C_1 &= aC_{\alpha f} - bC_{\alpha r} \\
C_2 &= aC_{\alpha f}^2 + bC_{\alpha r}^2 \\
I_{eq} &= I_x + mh^2
\end{aligned}
\tag{2}
$$

**Table 1.** Description of vehicle parameters.

| Parameter | Value | Description |
|---|---|---|
| $a$ | 1.42 m | Distance of the front axle from the center of gravity (CoG) |
| $b$ | 0.85 m | Distance of the rear axle from the CoG |
| $k_r$ | 31,752 N m/rad | Roll stiffness |
| $b_r$ | 7025.4 N m s/rad | Roll damping coefficient |
| $C_{\alpha f}$ | 30,000 N/rad | Cornering stiffness of the front tire |
| $C_{\alpha r}$ | 25,000 N/rad | Cornering stiffness of the rear tire |
| $g$ | 9.81 m/s$^2$ | Acceleration of gravity |
| $h$ | 0.35 m | Distance from roll center to CoG |
| $I_x$ | 520 kg m$^2$ | Moment of inertia about the roll axis |
| $I_z$ | 1110.9 kg m$^2$ | Moment of inertia about the yaw axis |
| $m$ | 650 kg | Vehicle mass |
| $v_x$ | m/s | Longitudinal speed |

Since the vehicle behavior is non-linear, a linear parameter varying (LPV)-based state-space vehicle model was adopted for this work, as it is more reliable when used with a linear model. For this particular case, the definition of a LPV-based model allows the variability of the state-space matrices to be considered due to the change in the vehicle velocity. Considering the significant variations in vehicle parameters, such as tire cornering stiffness, which are challenging to measure directly, model uncertainties were incorporated into the analysis. Equation (1) is rewritten in the state-space form:

$$
\begin{aligned}
\dot{x}(t) = {} & (A(\mu) + \Delta A(\mu))x(t) + (B_\delta(\mu) + \Delta B_\delta(\mu))\delta(t) \\
& + (B_u(\mu) + \Delta B_u(\mu))u(t) + B_\omega(\mu)w(t)
\end{aligned}
\tag{3}
$$

where $x = \begin{bmatrix} \beta & r & \phi & \dot{\phi} \end{bmatrix}^\top$ is the system state vector, which includes the sideslip angle, $\beta$, yaw rate, $r$, roll angle, $\phi$, and roll rate, $\dot{\phi}$. $\delta$ defines the steering wheel angle, and the control input is denoted by $u = \begin{bmatrix} \Delta\delta & M_\phi \end{bmatrix}^\top$. The disturbance is given by $w = \begin{bmatrix} \phi_r & \dot{p}_f & d \end{bmatrix}^\top$, with $\phi_r$ as the road bank angle, $\dot{p}_f$ as the longitudinal component of the angular velocity vector of the vehicle with respect to the inertial coordinates, and $d$ as the system noise.

The system polytope depends on the time-varying vector defined as $\mu = \begin{bmatrix} 1/v_x & 1/v_x^2 \end{bmatrix}^\top$. The matrices of model (3) are

$$
A(\mu) = \begin{bmatrix}
-\frac{I_{eq}(C_{\alpha f}+C_{\alpha r})}{I_x m v_x} & -\left(1 + \frac{I_{eq}(aC_{\alpha f}-bC_{\alpha r})}{I_x m v_x^2}\right) & \frac{h(mgh-k_r)}{I_x v_x} & -\frac{hb_r}{I_x v_x} \\
-\frac{(aC_{\alpha f}-bC_{\alpha r})}{I_z} & -\frac{(a^2 C_{\alpha f}+b^2 C_{\alpha r})}{I_z v_x} & 0 & 0 \\
0 & 0 & 0 & 1 \\
-\frac{(C_{\alpha f}+C_{\alpha r})h}{I_x} & -\frac{(aC_{\alpha f}-bC_{\alpha r})h}{I_x v_x} & \frac{(mgh-k_r)}{I_x} & -\frac{b_r}{I_x}
\end{bmatrix}
$$

$$
\Delta A(\mu) = \begin{bmatrix}
-\frac{I_{eq}(\Delta C_{\alpha f}+\Delta C_{\alpha r})}{I_x m v_x} & -\left(\frac{I_{eq}(a\Delta C_{\alpha f}-b\Delta C_{\alpha r})}{I_x m v_x^2}\right) & 0 & 0 \\
-\frac{(a\Delta C_{\alpha f}-b\Delta C_{\alpha r})}{I_z} & -\frac{(a^2 \Delta C_{\alpha f}+b^2 \Delta C_{\alpha r})}{I_z v_x} & 0 & 0 \\
0 & 0 & 0 & 1 \\
-\frac{(\Delta C_{\alpha f}+\Delta C_{\alpha r})h}{I_x} & -\frac{(a\Delta C_{\alpha f}-b\Delta C_{\alpha r})h}{I_x v_x} & 0 & 0
\end{bmatrix}
$$

$$
B_\delta(\mu) = \begin{bmatrix} \frac{I_{eq}C_{\alpha f}}{I_x m v_x} \\ \frac{aC_{\alpha f}}{I_z} \\ 0 \\ \frac{C_{\alpha f}h}{I_x} \end{bmatrix},\;
\Delta B_\delta(\mu) = \begin{bmatrix} \frac{I_{eq}\Delta C_{\alpha f}}{I_x m v_x} \\ \frac{a\Delta C_{\alpha f}}{I_z} \\ 0 \\ \frac{\Delta C_{\alpha f}h}{I_x} \end{bmatrix},\;
B_u(\mu) = \begin{bmatrix} \frac{I_{eq}C_{\alpha f}}{I_x m v_x} & 0 \\ \frac{aC_{\alpha f}}{I_z} & 0 \\ 0 & 0 \\ \frac{C_{\alpha f}h}{I_x} & \frac{1}{I_x} \end{bmatrix},\;
\Delta B_u(\mu) = \begin{bmatrix} \frac{I_{eq}\Delta C_{\alpha f}}{I_x m v_x} & 0 \\ \frac{a\Delta C_{\alpha f}}{I_z} & 0 \\ 0 & 0 \\ \frac{\Delta C_{\alpha f}h}{I_x} & 0 \end{bmatrix}
$$

$$
B_\omega(\mu) = \begin{bmatrix} -\frac{g}{v_x} & 0 & 1 \\ 0 & 0 & 1 \\ 0 & 0 & 1 \\ 0 & -1 & 1 \end{bmatrix}
$$

where $A$, $B_\delta$, $B_u$, and $B_\omega$ represent the nominal behavior of the state-space model, while $\Delta A$, $\Delta B_\delta$, and $\Delta B_u$ denote the cornering stiffness uncertainty effect.

The sideslip and roll angles are estimated through an observer with the following structure:

$$
\begin{aligned}
\dot{\tilde{x}} &= \sum_{i=1}^{3} \alpha_i [A_i \tilde{x} + B_{i,\delta} \delta + B_{i,u} u + L_i(\bar{y} - \tilde{y})] \\
\tilde{y} &= C \tilde{x} \\
\bar{y} &= e_y + y_{t-\eta}
\end{aligned}
\tag{4}
$$

where $L_i$ is the observation gain matrix to be designed and $\eta$ is the time delay due to the sampling and data transmission in the network. By defining the state error vector as $e_x = x - \tilde{x}$, and a new state vector $\xi = \begin{bmatrix} e_x & x \end{bmatrix}^T$, the combination (3) and (4) leads to

$$
\dot{\xi} = \sum_{i=1}^{3} \alpha_i [(A_{0,i} + \Delta A_{0,i})\xi + A_{1,i}\xi_{t-\eta} + A_{2,i}e_y + (A_{3,i} + \Delta A_{3,i})q]
\tag{5}
$$

where $q = \begin{bmatrix} \delta & u & w \end{bmatrix}^T$. The matrices in (5) are

$$
A_{0,i} = \begin{bmatrix} (A_i - L_iC) & L_iC \\ 0 & A_i \end{bmatrix}, \quad A_{1,i} = \begin{bmatrix} 0 & L_iC \\ 0 & 0 \end{bmatrix}, \quad A_{2,i} = \begin{bmatrix} -L_i \\ 0 \end{bmatrix}, \quad A_{3,i} = \begin{bmatrix} 0 & 0 & B_{\omega,i} \\ B_{\sigma,i} & B_{u,i} & B_{\omega,i} \end{bmatrix}
$$

$$
\Delta A_{0,i} = \begin{bmatrix} 0 & \Delta A_i \\ 0 & \Delta A_i \end{bmatrix}, \quad \Delta A_{3,i} = \begin{bmatrix} \Delta B_{\delta,i} & \Delta B_{u,i} & 0 \\ \Delta B_{\delta,i} & \Delta B_{u,i} & 0 \end{bmatrix}
$$

The problem of the event-triggered $\mathscr{H}_\infty$ observer design is based on finding the observer gains, $L_i$, such that:

- When $q(t) = 0$, the system (5) is asymptotically stable;
- Under the zero initial condition, $||z^\top z||_2 < \gamma^2 ||q^\top q||_2$ holds for $q(t) \neq 0 \in \mathcal{L}_2\,[0, \infty)$, with $\gamma$ being the $\mathscr{H}_\infty$ performance index.

The observation gain matrices, $L_i$, are obtained by solving the following linear matrix inequality (LMI) problem:

**Theorem 1.** *The event-triggered observer presented in (4) is asymptotically stable with an $\mathscr{H}_\infty$ performance index $\gamma > 0$ if, for given values of $\eta_m > 0$, $\eta_M > \eta_m$, $h > 0$, $\epsilon > 0$, $\rho_1 > 0$ and $\rho_2 > 0$, there exist matrices $P_1 = P_1^\top > 0$, $P_2 = P_2^\top > 0$, $T_{1,i} = T_{1,i}^\top > 0$, $T_{2,i} = T_{2,i}^\top > 0$, $S_1 = S_1^\top > 0$, $S_2 = S_2^\top > 0$, $\Omega_i = \Omega_i^\top > 0$ and matrices $M_i$ and $R$ of appropriate dimensions, such that*

$$
\begin{bmatrix} \overline{\Sigma}_{11,i} & \overline{\Sigma}_{12,i} & \overline{\Sigma}_{13,i} \\ \star & \overline{\Sigma}_{22,i} & \overline{\Sigma}_{23,i} \\ \star & \star & \overline{\Sigma}_{33,i} \end{bmatrix} < 0, \quad for\ i = 1, \ldots, 3
$$

$$
\begin{bmatrix} S_2 & R \\ \star & S_2 \end{bmatrix} > 0
\tag{6}
$$

*where*

$$P = \begin{bmatrix} P_1 & 0 \\ 0 & P_2 \end{bmatrix}, \ \overline{A}_{0,i} = \begin{bmatrix} P_1 A_i - M_i C & M_i C \\ 0 & P_2 A_i \end{bmatrix}, \ \overline{A}_{1,i} = \begin{bmatrix} 0 & -M_i C \\ 0 & 0 \end{bmatrix}, \ \overline{A}_{2,i} = \begin{bmatrix} -M_i \\ 0 \end{bmatrix}$$

$$\overline{A}_{3,i} = \begin{bmatrix} 0 & 0 & P_1 B_{\omega,i} \\ P_2 B_{i,\delta} & P_2 B_{i,u} & P_2 B_{i,\omega} \end{bmatrix}, \ \hat{E}_{A,i} = \begin{bmatrix} 0 & P_1 E_{A,i} \\ 0 & P_2 E_{A,i} \end{bmatrix}, \ \hat{E}_{B,i} = \begin{bmatrix} P_1 E_{B_{\delta},i} & P_1 E_{B_{u},i} & 0 \\ P_2 E_{B_{\delta},i} & P_2 E_{B_{u},i} & 0 \end{bmatrix}$$

$$\overline{\Sigma}_{11,1} = \begin{bmatrix} \overline{\Sigma}_{11,i,1,1} & \overline{A}_{1,i} & S_1 & 0 \\ \star & \overline{\Sigma}_{11,i,2,2} & S_2 + R & S_2 + R^\top \\ \star & \star & T_{2,i} - T_{1,i} - S_1 - S_2 & -R^\top \\ \star & \star & \star & -T_{2,i} - S_2 \end{bmatrix}$$

$$\overline{\Sigma}_{11,i,1,1} = \overline{A}_{0,i}^\top + \overline{A}_{0,i} + T_{1,i} - S_1, \ \overline{\Sigma}_{11,i,2,2} = -2S_2 - R^\top - R + \varepsilon \Theta^\top \Omega_i \Theta$$

$$\overline{\Sigma}_{12,i} = \begin{bmatrix} \overline{A}_{2,i} & \overline{A}_{3,i} & C_1^\top & \eta_m \overline{A}_{0,i}^\top & (\overline{\eta} - \eta_m) \overline{A}_{0,i}^\top \\ \varepsilon \Theta^\top \Omega_i & 0 & 0 & \eta_m \overline{A}_{1,i}^\top & (\overline{\eta} - \eta_m) \overline{A}_{1,i}^\top \\ 0 & 0 & 0 & 0 & 0 \\ 0 & 0 & 0 & 0 & 0 \end{bmatrix}, \ \overline{\Sigma}_{13,i} = \begin{bmatrix} \hat{E}_{A,i} & \mu_1 \overline{F}_A^\top & \hat{E}_{B,i} & 0 \\ 0 & 0 & 0 & 0 \\ 0 & 0 & 0 & 0 \\ 0 & 0 & 0 & 0 \end{bmatrix}$$

$$\overline{\Sigma}_{22,i} = \begin{bmatrix} \Omega_i + \epsilon \Omega_i & 0 & 0 & \tau_m \overline{A}_{2,i}^\top & (\overline{\eta} - \tau_m) \overline{A}_{2,i}^\top \\ \star & -\gamma^2 I & 0 & \tau_m \overline{A}_{3,i}^\top & (\overline{\eta} - \tau_m) \overline{A}_{3,i}^\top \\ \star & \star & -I & 0 & 0 \\ \star & \star & \star & \rho_1^2 S_1 - 2\rho_1 P & 0 \\ \star & \star & \star & \star & \rho_2^2 S_2 - 2\rho_2 P \end{bmatrix}$$

$$\overline{\Sigma}_{23,i} = \begin{bmatrix} 0 & 0 & 0 & 0 \\ 0 & 0 & 0 & \mu_2 \overline{F}_B^\top \\ 0 & 0 & 0 & 0 \\ \eta_m \hat{E}_{A,i} & 0 & \eta_m \hat{E}_{B,i} & 0 \\ (\overline{\eta} - \eta_m) \hat{E}_{A,i} & 0 & (\overline{\eta} - \eta_m) \hat{E}_{B,i} & 0 \end{bmatrix},$$

$$\overline{\Sigma}_{33,i} = \begin{bmatrix} -\mu_1 I & 0 & 0 & 0 \\ \star & -\mu_1 I & 0 & 0 \\ \star & \star & -\mu_2 & 0 \\ \star & \star & \star & -\mu_2 I \end{bmatrix}$$

$$\tag{7}$$

*Once a feasible solution is found, the observation gain matrices are obtained by $L_i = P_1^{-1} M_i$.*

**Proof.** The proof is presented in [24] and omitted here for simplicity. □

### 2.3. Software Perspective

As Figure 1 shows, cloud services are divided into two components: A REST API, which implements an interface to store and query the data from the vehicle, and a Database Manager, which is used by the REST API to store vehicle data in a persistent manner.

The vehicle localization and estimated data are sent to the REST API and then a web-based client queries these data, asking the REST API via HTTP petitions. This web-based client is employed to display a map with geo-located data of the vehicle in real time.

Figure 2 presents the component diagram of the system, where each dependency among components is easily visualized. In order to keep the component diagram as simple as possible, only one interface is shown for each component. However, in the real architecture, a component may provide implementation for a wide range of functions.

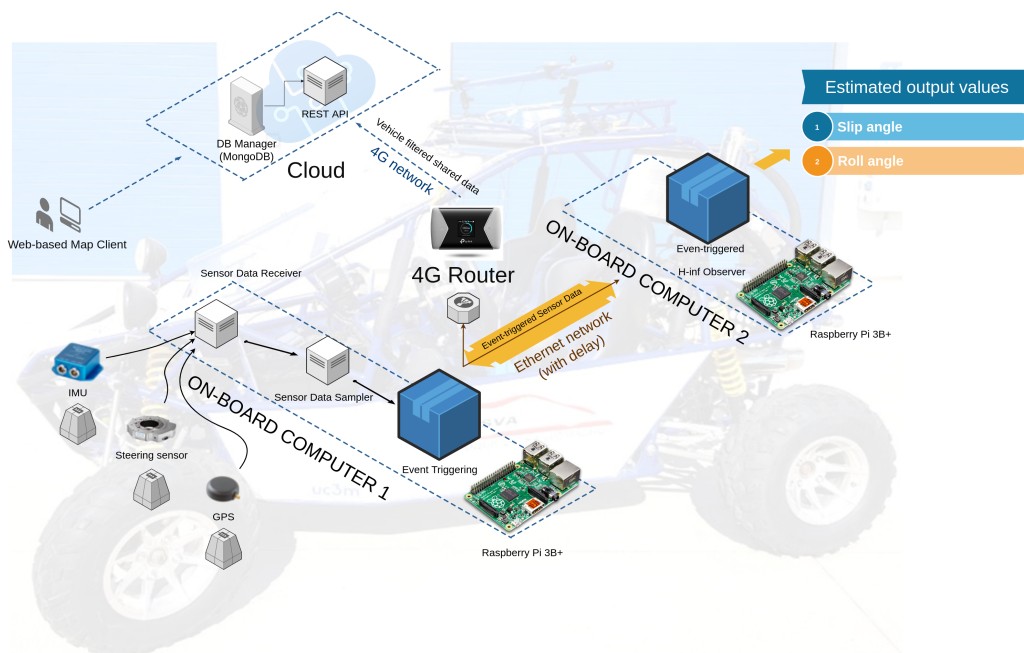

**Figure 1.** IoT-based architecture applied to the event-triggered observer.

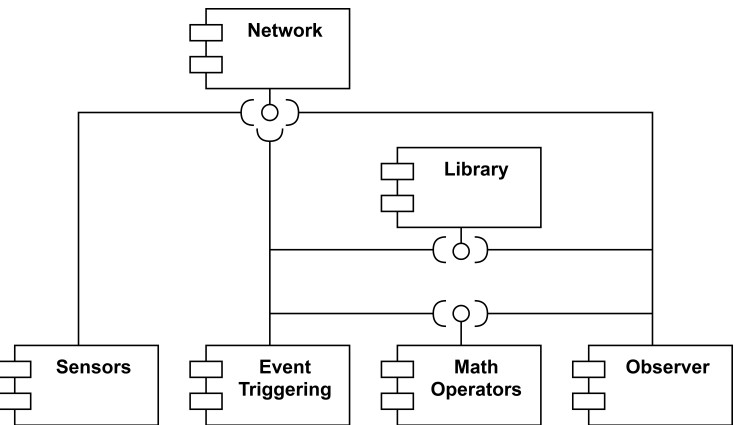

**Figure 2.** Component diagram.

As depicted in Figure 2, the system under consideration comprises multiple components, each designed with a specific function. The most relevant are as follows:

a.  Network. This component incorporates the essential logic required to facilitate communication among the computers within the system. At the time of writing this article, the system employed a client–server model utilizing TCP sockets for data exchange. This component furnishes various implementations for distinct operations, including:

- Establish communication: (a) receives the IP and port of the machine; and (b) returns a socket descriptor that is used to exchange data with the process;
- Set socket listener port: (a) receives the port number in which the server will be listening for new connections; and (b) returns a number that will be used by the server as socket descriptor;
- Send message: (a) receives the message to send, the length of the message, and one socket descriptor; and (b) returns 0 in case of success and $-1$ if an error occurs;
- Receive message; (a) receives a buffer where the message will be stored, the length of the message, and one socket descriptor; and (b) returns 0 in case of success and $-1$ if an error occurs.

b.  Math operators. The programs in development must execute various mathematical operations, including matrix calculations. The responsibility of this block is to ensure the efficient implementation of these operators. Among the functions offered by this component are:

- Create matrix;
- Multiply matrix by number;
- Add matrices;
- Multiply matrices;
- Transpose matrix;
- Convert degrees to radians.

c.  Sensors. This component represents the array of sensors utilized by the Raspberry Pi for environmental data collection. To enhance the system modularity, it was determined that this component should employ the network interface for data transmission instead of being integrated into the event-triggering block. This choice enables developers to separate the data collection process from its subsequent processing.

d.  Event triggering. This block executes the event-triggering algorithm. It processes data from sensors and transmits that information to the observer. Given that the algorithm involves matrix operations, it is imperative to ensure the availability and integration of the math component for its proper functionality.

e.  Observer. This component is the last piece of the algorithm. It processes the data sent by the event-triggering program and estimates both the vehicle sideslip and roll angles.

Note: Both the event-triggering program and the observer are required not only to communicate with each other and perform mathematical operations, but also to generate graphics once the programs have finished. This is why they are connected to the library component.

The algorithm presented in Figure 3 illustrates the implementation of the proposed system, which comprises three separate programs. To ensure the generalizability, it was assumed that the vehicle is not in an infinite loop, although it may complete the algorithm if necessary. Consequently, the algorithm includes several decision blocks that assess whether the programs should terminate in the current iteration. In order to model concurrency between programs, the diagram employs fork and join. It is essential to note that, in the actual project, the user activates distinct codes independently. A new thread is not created by a program to initiate the execution of another program. Before continuing, it is important to note that the diagram was created under the assumption that the programs would be written in C++. As a result, there are several blocks responsible for freeing up resources such as memory that were reserved during the iteration. However, this diagram can be used to explain any other implementation in a different language by simply disregarding those blocks.

- Main: this program is responsible for obtaining data from the sensors and transmitting them to the event-triggering program;
- Event triggering: this code receives data from the sensors and performs several mathematical operations in order to decide whether the processed information must be sent to the observer or not;
- Observer: this program calculates the slip and roll angles as its final outputs.

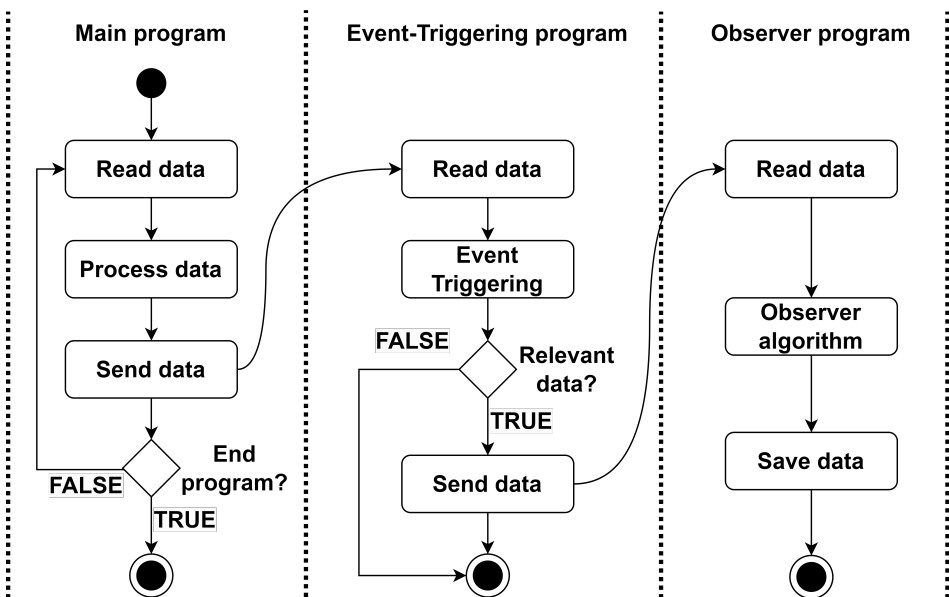

**Figure 3.** Activity diagram.

### 2.4. Experimental Design

The experimental tests were carried out in the National Institute for Aerospace Technology (INTA) testing tracks located in Madrid, Spain, (see Figure 4) using the testbed configuration described in Section 2.1. To evaluate the hypothesis defined for this research, the following experiments were executed:

a.   Spiral. During a spiral movement, the driver steers the vehicle along a path that gradually narrows or widens in a spiral shape. The challenge is to maintain control and follow the curvature of the road.

b.   Slalom. The slalom is a driving course that consists of a series of gates or markers arranged in a zigzag pattern. The driver must navigate through the obstacles by making rapid turns and quick changes in direction.

c.   Lane change. The ability to change lanes is a crucial driving skill that has many practical applications in everyday driving scenarios. For example, changing lanes allows overtaking slower vehicles, adjusting to traffic conditions, and avoiding obstacles.

These maneuvers are particularly valuable for evaluating a vehicle's dynamic capabilities, including its ability to change direction rapidly and handle tight turns. Through the execution of these experiments, an accurate approximation of the nominal vehicle parameters was obtained (see Table 1). Accurately determining these parameters is crucial for creating reliable observers for the sideslip and roll angle.

### 2.5. Data Gathering

The data obtained for each of the previously defined experiments (see Section 2.4) were stored by the controller of each kit in a CSV-formatted file, showing the experiment and its execution date and time. The stored variables were as follows:

- Covariance between pitch and roll angles;
- Covariance between yaw and pitch angles;
- Covariance between yaw and roll angles;
- Altitude;
- Latitude;
- Longitude;
- Pitch angle;
- Pitch rate;
- Roll angle;
- Roll rate;

- Yaw angle;
- Yaw rate;
- Vehicle speed;
- Timestamp.

These measurements were gathered according to the sampling rate stated for the experiments, which was 100 Hz. Appendix A presents an example of the datasets obtained from the experiments.

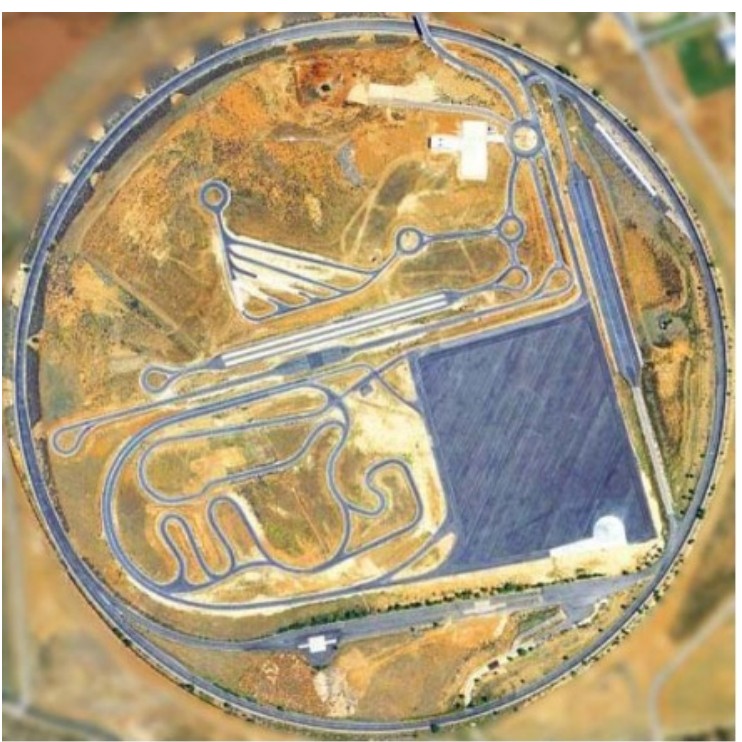

**Figure 4.** Map of the experimental test location.

## 3. Experimental Results

To validate the feasibility and performance of the proposed architecture for estimating roll and sideslip angles, experimental tests were carried using a GOKA 650 vehicle (see Figure 5).

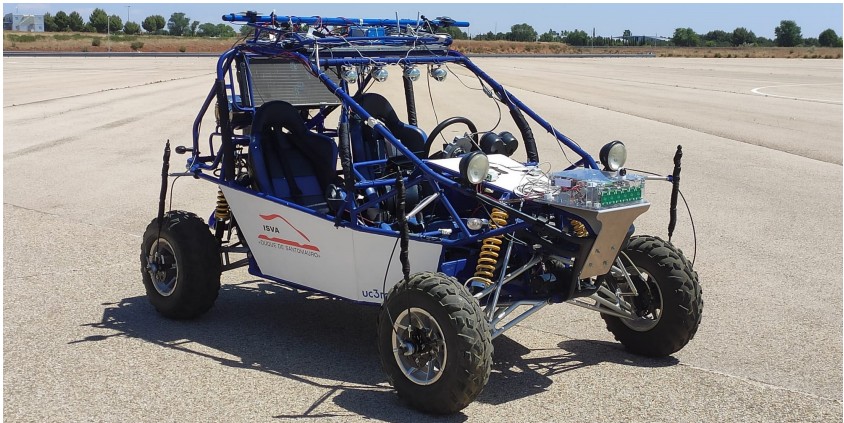

**Figure 5.** Goka 650 vehicle used for the experiment.

Vehicle parameters for the model (1) were obtained after performing several spiral and lane change experiments with the real vehicle (see Table 1). An accurate model for the

real vehicle lateral dynamic behavior is required in order to design a robust observer that is capable of estimating both the roll and sideslip angles.

The event-triggered-based IoT observer was designed under the following considerations:

- It was assumed that the longitudinal velocity was bounded in the interval [2 m/s, 20 m/s];
- The maximum variation in the tire cornering stiffness was 5% of its nominal value;
- The data sampling frequency was 100 Hz, with minimum and maximum transmission time delays of 5 and 20 ms, respectively.

A slalom test was performed to evaluate the effectiveness of the proposed architecture. This maneuver was chosen for validation since, in this, the vehicle's dynamic capabilities are more stressed than in the other tests. The path that the vehicle followed during the experiment is depicted in Figure 6. The steering wheel angle set by the driver is presented in Figure 7. The longitudinal velocity profile of the vehicle during this experiment is shown in Figure 8.

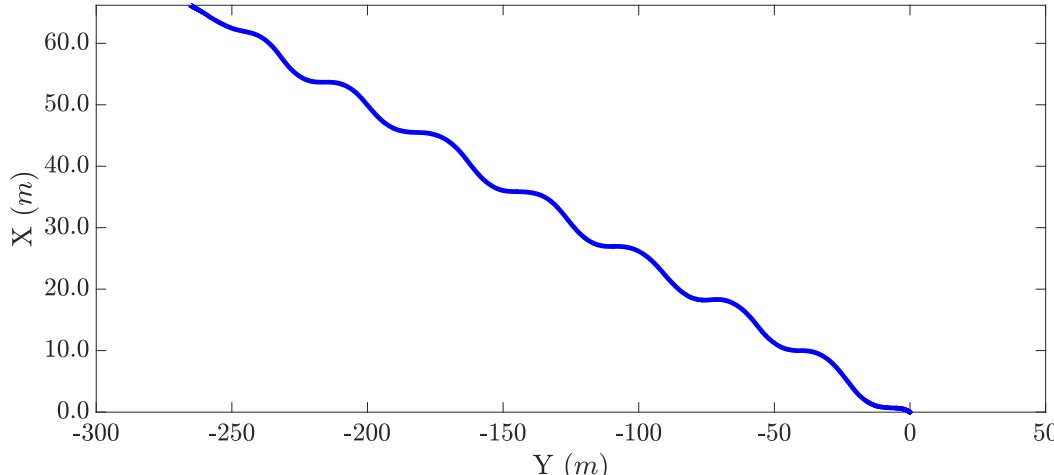

**Figure 6.** Path followed in the experiment.

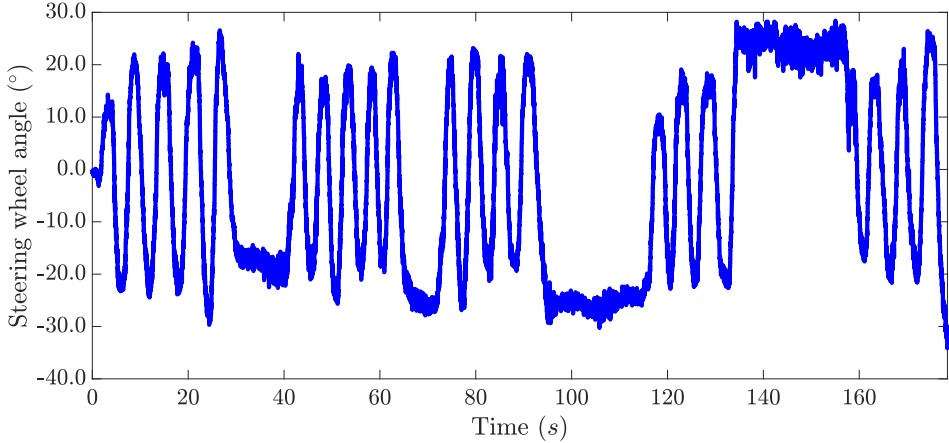

**Figure 7.** Steering wheel angle during the experiment.

Figure 9 shows the real and estimated sideslip angle of the vehicle during the experiment. Figure 10 depicts the comparison between the real and estimated roll angle of the vehicle. The ground truth values of the sideslip and roll angle were obtained using a dual GPS antenna, which is a high-cost sensor. Therefore, it is not possible to consider this as part of a standard vehicle.

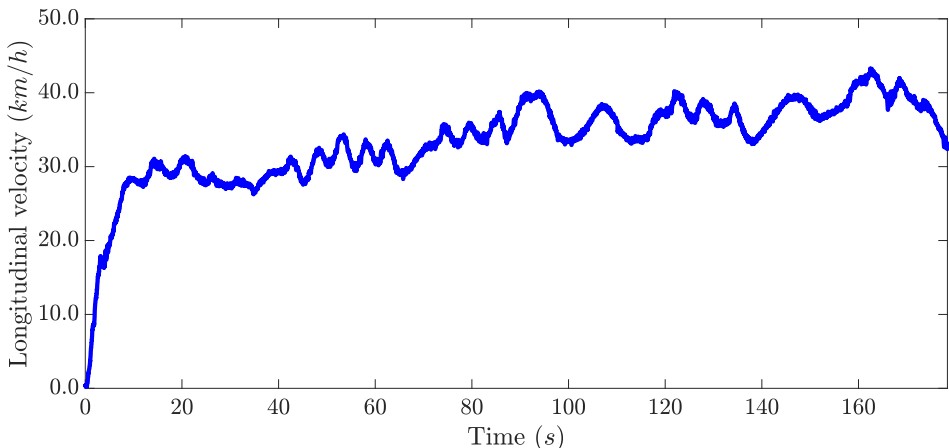

**Figure 8.** Velocity of the vehicle during the experiment.

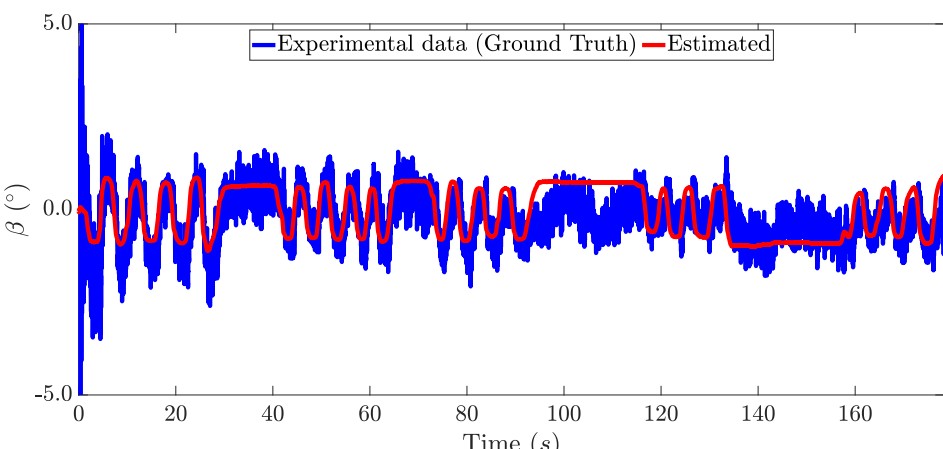

**Figure 9.** Sideslip angle estimation.

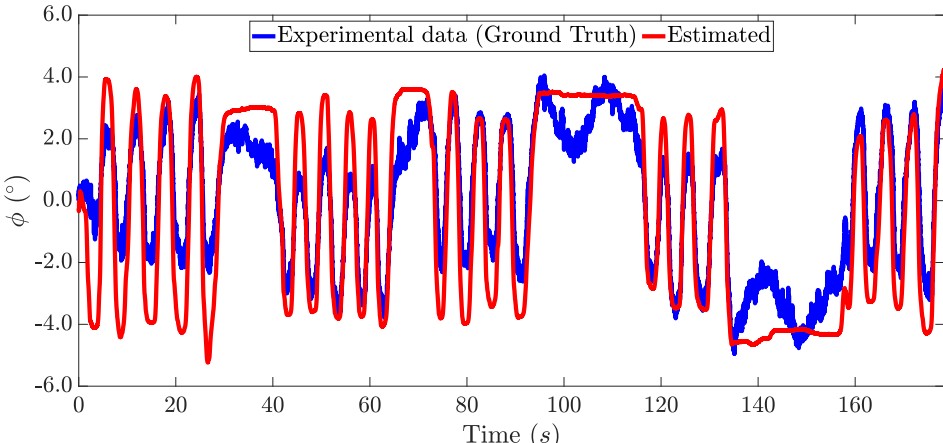

**Figure 10.** Roll angle estimation.

**Table 2.** Estimation errors obtained in the experimental test.

|  | Sideslip Angle ($\beta$) | Roll Angle ($\phi$) |
| --- | --- | --- |
| *MAX error* | 2.11° | 3.44° |
| *RMS error* | 0.28° | 0.25° |

In order to quantify the experimental results, Table 2 shows the maximum absolute error (MAX error) and the root mean square error (RMS error) for the estimated data, which were calculated as follows:

$$MAX\ error\ \beta = MAX(|\beta_{experimental} - \beta_{estimated}|)$$
$$MAX\ error\ \phi = MAX(|\phi_{experimental} - \phi_{estimated}|)$$
$$RMS\ error\ \beta = \sqrt{\frac{1}{n}\sum_{i=1}^{n}(\beta_{experimental,i} - \beta_{estimated,i})^2} \quad (8)$$
$$RMS\ error\ \phi = \sqrt{\frac{1}{n}\sum_{i=1}^{n}(\phi_{experimental,i} - \phi_{estimated,i})^2}$$

The RMS error indicates a difference between the real and estimated value lower than $0.3°$, for both the roll and sideslip angles. Thus, this demonstrates that the proposed architecture produced a good estimation for these unmeasured angles. With respect to the use of an event-triggered condition, a non-transmission rate of 44.32% was achieved, which lowered the amount of data sent over the network and demonstrates the advantage of designing an observer with an event-triggering condition.

Regarding the hypothesis introduced in Section 2, it was verified that the proposed IoT system can approximate vehicle roll and sideslip angles in real time without overloading the intra-vehicle network.

## 4. Conclusions

An event-triggered-based IoT architecture that simultaneously estimates the sideslip angle, $\beta$, and the roll angle, $\phi$, is proposed in this paper. The observer was designed considering network-induced delays and an event-triggering mechanism to reduce the intra-vehicle network load.

The experimental results demonstrate that the proposed observer can simultaneously estimate both roll and sideslip angles using information from sensors already available in series-production vehicles or low-cost sensors, such as the roll rate, $\dot{\phi}$, yaw rate, $r$, and longitudinal velocity, $v_x$. This ensures that the cost of the proposed architecture remains low.

The incorporation of an event-triggering mechanism allows the system to neglect any data package with irrelevant information, which reduces the amount of information sent over the network and, thus, the chance of the network becoming saturated. As shown in the experimental results, a non-transmission rate of 44.32% was achieved.

In future work, different event-triggering mechanisms will be studied. Furthermore, we will compare the observer in this work with other alternative observers such as the Kalman filter. Additionally, the system for estimating roll and sideslip angles herein can be used in designing rollover prevention systems using semi-active suspension and skid prevention systems through torque vectoring techniques, such as differential braking.

**Author Contributions:** F.V.-M., J.G., M.J.L.B. and B.L.B. proposed the ideas; F.V.-M., M.J.L.B. and B.L.B. performed the mathematical development; F.V.-M., J.G., M.M.-U. and M.J.-S. conceived and designed the experiments; F.V.-M., J.G., M.J.L.B. and B.L.B. analyzed the data; F.V.-M., J.G., M.M.-U., M.J.-S., M.J.L.B. and B.L.B. wrote the paper. All authors have read and agreed to the published version of the manuscript.

**Funding:** Grant [ PID2022-136468OB-I00 ] funded by MCIN/AEI/ 10.13039/501100011033 and by "ERDF A way of making Europe".

**Data Availability Statement:** The original contributions presented in the study are included in the article, further inquiries can be directed to the corresponding author.

**Conflicts of Interest:** The authors declare no conflicts of interest.

## Abbreviations

The following abbreviations are used in this manuscript:

ADS    Automated driving system
GPS    Global positioning system
NCS    Networked control system
IMU    Inertial measurement unit
IoT    Internet of Things
LPV    Linear parameter varying
RMS    Root mean square

## Appendix A. Example of Data Logged During the Experiments

| altitude | cov_pitch_roll | cov_yaw_pitch | cov_yaw_pitch | latitude | longitude | pitch | pitch_rate | roll | roll_rate | speed | timestamp | yaw | yaw_rate |
|---|---|---|---|---|---|---|---|---|---|---|---|---|---|
| 622.8 | 0 | 0 | 0 | 0 | 0 | -6.8125 | -0.0021817 | -6.1875 | -0.0021817 | 0 | 1656912149 | 360 | -0.0021817 |
| 622.8 | 0 | 0 | 0 | 0 | 0 | -6.8125 | -0.0032725 | -6.1875 | -0.0032725 | 0 | 1656912149 | 359.9375 | -0.0010908 |
| 622.8 | 0 | 0 | 0 | 40.899 | -3.29 | -6.8125 | 0 | -6.1875 | 0 | 0.0051 | 1656912149 | 360 | 0.00109083 |
| 622.8 | 0 | 0 | 0 | 40.899 | -3.29 | -6.8125 | -0.0010908 | -6.1875 | -0.0010908 | 0.0051 | 1656912150 | 359.9375 | 0.00109083 |
| 622.8 | 0 | 0 | 0 | 40.899 | -3.29 | -6.8125 | 0 | -6.1875 | 0 | 0.0051 | 1656912150 | 359.9375 | -0.0010908 |
| 622.8 | 0 | 0 | 0 | 40.899 | -3.29 | -6.8125 | 0.00109083 | -6.1875 | 0.00109083 | 0.0051 | 1656912150 | 359.9375 | 0.00218166 |
| 622.8 | 0 | 0 | 0 | 40.899 | -3.29 | -6.8125 | -0.0021817 | -6.1875 | -0.0021817 | 0 | 1656912150 | 359.9375 | 0 |
| 622.8 | 0 | 0 | 0 | 40.899 | -3.29 | -6.8125 | -0.0010908 | -6.1875 | -0.0010908 | 0 | 1656912150 | 359.9375 | 0.00218166 |
| 622.8 | 0 | 0 | 0 | 40.899 | -3.29 | -6.8125 | 0.00218166 | -6.1875 | 0.00218166 | 0 | 1656912150 | 359.9375 | 0.00218166 |
| 622.8 | 0 | 0 | 0 | 40.899 | -3.29 | -6.8125 | 0 | -6.1875 | 0 | 0 | 1656912150 | 359.9375 | 0 |
| 622.8 | 0 | 0 | 0 | 40.899 | -3.29 | -6.8125 | 0.00218166 | -6.1875 | 0.00218166 | 0 | 1656912150 | 359.9375 | -0.0010908 |
| 622.8 | 0 | 0 | 0 | 40.899 | -3.29 | -6.8125 | -0.0021817 | -6.1875 | -0.0021817 | 0 | 1656912151 | 359.9375 | -0.0010908 |
| 622.8 | 0 | 0 | 0 | 40.899 | -3.29 | -6.8125 | 0.00109083 | -6.1875 | 0.00109083 | 0 | 1656912151 | 359.9375 | 0.00109083 |
| 622.8 | 0 | 0 | 0 | 40.899 | -3.29 | -6.8125 | 0.00109083 | -6.1875 | 0.00109083 | 0 | 1656912151 | 359.9375 | 0.00109083 |
| 622.8 | 0 | 0 | 0 | 40.899 | -3.29 | -6.8125 | -0.0010908 | -6.1875 | -0.0010908 | 0.0051 | 1656912151 | 359.9375 | 0.00109083 |
| 622.8 | 0 | 0 | 0 | 40.899 | -3.29 | -6.8125 | 0.00218166 | -6.1875 | 0.00109083 | 0.0051 | 1656912151 | 359.9375 | 0.00109083 |
| 622.8 | 0 | 0 | 0 | 40.899 | -3.29 | -6.8125 | 0 | -6.1875 | -0.0010908 | 0.0051 | 1656912151 | 359.9375 | -0.0021817 |
| 622.8 | 0 | 0 | 0 | 40.899 | -3.29 | -6.8125 | -0.0021817 | -6.1875 | -0.0021817 | 0.0051 | 1656912151 | 359.9375 | -0.0021817 |
| 622.8 | 0 | 0 | 0 | 40.899 | -3.29 | -6.8125 | -0.0021817 | -6.1875 | -0.0021817 | 0 | 1656912152 | 359.9375 | -0.0010908 |
| 622.8 | 0 | 0 | 0 | 40.899 | -3.29 | -6.8125 | 0.00109083 | -6.1875 | 0.00109083 | 0 | 1656912152 | 359.9375 | -0.0021817 |
| 622.8 | 0 | 0 | 0 | 40.899 | -3.29 | -6.8125 | -0.0021817 | -6.1875 | -0.0021817 | 0 | 1656912152 | 359.9375 | 0.00218166 |
| 622.8 | 0 | 0 | 0 | 40.899 | -3.29 | -6.8125 | 0 | -6.1875 | 0 | 0 | 1656912152 | 359.9375 | 0.00109083 |
| 622.8 | 0 | 0 | 0 | 40.899 | -3.29 | -6.8125 | 0.00218166 | -6.1875 | 0.00218166 | 0 | 1656912152 | 359.9375 | -0.0010908 |
| 622.8 | 0 | 0 | 0 | 40.899 | -3.29 | -6.8125 | 0 | -6.1875 | 0.00218166 | 0 | 1656912152 | 359.9375 | 0.00436332 |
| 622.8 | 0 | 0 | 0 | 40.899 | -3.29 | -6.8125 | 0.00109083 | -6.1875 | 0 | 0 | 1656912152 | 359.9375 | 0 |
| 622.8 | 0 | 0 | 0 | 40.899 | -3.29 | -6.8125 | 0.00436332 | -6.1875 | 0.00109083 | 0 | 1656912152 | 359.9375 | -0.0010908 |
| 622.8 | 0 | 0 | 0 | 40.899 | -3.29 | -6.8125 | 0 | -6.1875 | 0.00436332 | 0.0051 | 1656912153 | 359.9375 | -0.0010908 |

**Figure A1.** CSV file contents showcasing data logged by the VBOX kit throughout the course of the experiments.

```json
{
    "altitude": "626.6",
    "cov_pitch_roll": 0,
    "cov_yaw_pitch": 0,
    "cov_yaw_roll": 0,
    "latitude": 40.907,
    "longitude": -3.247,
    "pitch": 0.25,
    "pitch_rate": -0.006544984694978736,
    "roll": 0.4375,
    "roll_rate": -0.006544984694978736,
    "speed": 0.0,
    "temperature": 46,
    "timestamp": 1656912676,
    "vehicle": 1,
    "yaw": 0.0625,
    "yaw_rate": 0.00545415391248228
},
```

**Figure A2.** JSON file contents showcasing data recorded by low-cost sensors throughout the execution of experiments.

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
