# Peer review of "Simultaneous Estimation of Vehicle Sideslip and Roll Angles Using an Event-Triggered-Based IoT Architecture"

_machines, doi:10.3390/machines12010053_

Round 1

Reviewer 1 Report

Comments and Suggestions for Authors

This is an interesting paper with suitable content for this journal. The following comments are recommended to the authors:

1_ Please better justify the selected applications for a smart anti-slip assessment. Please consult and probably cite additional relevant references, e.g., https://doi.org/10.3390/vehicles4030042  

2_ Please better explain and give details about the experimental test setup presented in figure 4.

3_ The length of the paper and the discussion points in the result section are too short. Please expand the content.

4_ Table 2: What is the physical meaning and impact of the presented estimation errors.

5_ In the same context (as in point No. 3), please elaborate further on the conclusions, i.e., study limitations, research gaps and open challenges, future prospects should definitively be discussed.

Comments on the Quality of English Language

Moderate changes are needed.

Reviewer 2 Report

Comments and Suggestions for Authors

This paper presents experimental results on robust H-infinity observer for simultaneous estimation of side-slip and roll angles with an event-triggered-based IoT architecture. 

This manuscript seems not a paper but an implementation or experimental report. Without the theoretical contribution and validation, this has little contribution to be published in Machines. To have contribution, theoretical background should be presented. 

1. The authors should explain how to define and obtain Ai, Bi,delta and Bi,u in Eq. (1). Moreover, it should be explained that the physical meaning of Ai and Bi, and DeltaAi and DeltaBi

2. Brief procedure to calculate the observer gain Li should be presented in the last of the subsection 2.2. This is the main contribution, which is needed for pulication in Machines. More specifically, the observer design for a system with parameter uncertainty and network delay is a key contribution. So, design procedure of robust time-varying observer should be presented. 

3. Detailed specification on Racelogic VBOX, e.g., resoluton, sampling period and maximum rms error, was not given. Without this, it is hard to believe the accuracy of the ground truth values given in Figures 11 and 12. Moreover, the authors should present the detailed speficiation of the other sensors used for experiment. For example, the resolutions of RT3000 V3 of Oxford Technology are 0.03 deg for pitch and roll angles, and 0.15deg for side-slip angle.  

4. To validate the observer for a system with parameter uncertainty and network dealy, it should be compared with an observer or Kalman filter designed for a system without parameter uncertainty and network delay. Through this comparison, the advantages of the proposed structure or method can be validated. 

Comments on the Quality of English Language

N/A

Reviewer 3 Report

Comments and Suggestions for Authors

In this paper, the authors propose an event-triggered-based IoT architecture to estimate the sideslip and roll angle simultaneously. Through experiments, they also demonstrated that the structure has good performance and robustness when estimating the angle of these vehicle angles through low-cost devices. I think this work and some of the results presented are significant and interesting, and I would also like to encourage them to continue their research.

However, this work has some weakness, in my opinion, that require a revision:

1. There is large white space on page 5 and page 13, please pay attention to the layout of the pictures and charts on these pages.

2. Since the font in the figure on page 9 is too small to read, and there is a lot of white space above and below the figure.

3. The content in Figure 2 is blurry, please adjust its clarity.

4. Figures 5 and 6 present the data directly in the text, which is not in line with academic norms. Please put them in the appendix.

5. Figures 8, 9, 10, 11, and 12 are not properly formatted and the font size is too large.

6. In the part of "experimental results", the description of experimental results is too simple, just a large list of experimental charts. From your description, I cannot understand the intent of your experiment and what the final results confirm. For example, in lines 330 -331, please supplement in detail what hypothesis has been verified. Please describe the part of "experimental results" in detail.

Comments on the Quality of English Language

NO Comments

Round 2

Reviewer 1 Report

Comments and Suggestions for Authors

None.

Comments on the Quality of English Language

Minor checks are needed.

Reviewer 2 Report

Comments and Suggestions for Authors

All comments given in the first review process were resolved in the revised manuscript. I recommend this paper can be accepted to be published in Machines.